



# Measurement report: The first *in-situ* PM₁ chemical measurements at the steep slope from highly polluted Sichuan Basin to pristine Tibetan Plateau: light absorption of carbonaceous aerosols, and source and origin impacts

Suping Zhao[1,2,3,4], Shaofeng Qi[1,6], Ye Yu[1,2,3], Shichang Kang[4], Longxiang Dong[1,2,3], Jinbei Chen[1,2,3], Daiying Yin[5,6]

[1] Key Laboratory of Land Surface Process and Climate Change in Cold and Arid Regions, Northwest Institute of Eco-Environment and Resources, Chinese Academy of Sciences, Lanzhou 730000, China
[2] Pingliang Land Surface Process & Severe Weather Research Station, Pingliang, 744015, China
[3] Gansu Land Surface Process & Severe Weather Observation and Research Station, Pingliang, 744015, China
[4] State Key Laboratory of Cryospheric Science, Northwest Institute of Eco-Environment and Resources, Chinese Academy of Sciences, Lanzhou 730000, China
[5] Key Laboratory of Desert and Desertification, Northwest Institute of Eco-Environment and Resources, Chinese Academy of Sciences, Lanzhou 730000, China
[6] University of Chinese Academy of Sciences, Beijing 100049, China

*Correspondence to*: Suping Zhao (zhaosp@lzb.ac.cn); Daiying Yin (yindaiying@lzb.ac.cn)

**Abstract.** Tibetan Plateau (TP, hereafter), known as "Third Pole", is surrounded by the highly polluted regions, such as Indo-Gangetic Plain, Taklimakan and Gobi Deserts and Sichuan Basin. However, the previous *in-situ* aerosol measurements mainly focused on the southern and northern slopes, while less observations and studies were conducted at the eastern slope of the TP (ESTP). The scientific knowledge on optical properties of aerosols is extremely limited over the ESTP, and *in-situ* observations at varying altitudes from the heavily polluted regions to the relatively clean Plateau were important for better understanding the light absorption and radiative forcing over the TP. Sichuan Basin (SCB), a highly polluted region due to more rapid economic development, is located in the east side of TP. Therefore, we conducted the first aerosol field experiment at six sites (Chengdu, Sanbacun, Wenchuan, Lixian, Maerkang, Hongyuan) along eastern slope of the TP extending elevation from 500 m to 3500 m. Light-absorbing aerosols are considered to be a key climate driver, and their role may be underestimated in the high-altitude regions. The light absorption of brown carbon (BrC) accounting for



that of total carbon increases from 20% to 50%, and the mass absorption efficiency of BrC over TP is 2–3 times higher than that inside SCB, especially in winter, which is mainly related to high ratio of secondary to primary organic carbon due to stronger secondary formation and less primary emissions at high altitudes. Contrary to BrC aerosols, EC (elemental carbon) mass absorption efficiency declines

5    with altitude in winter, induced by source difference between the TP and SCB. The more urban sources (motor vehicles, industries, etc.) inside the SCB fail to be transported to the TP due to stable air inside the basin in winter, which also is favorable for aerosol aging to enhance absorption efficiency. The radiative forcing of BrC relative to EC varies from 0.10 to 0.42 as altitude increases with the higher OC/EC ratio over the TP than SCB, and thus the enhanced radiative forcing of BrC relative to EC from

10   polluted SCB to pristine TP is because the concentration of OC decreases more slowly with altitude than does EC. South Asia, a highly particulate matter (PM) pollution region, is an important origin of aerosol particles at the region from western Sichuan Basin to eastern Tibetan Plateau, which is significantly dependent on seasons.



## 1 Introduction

Some *in-situ* observations, available satellite data and model simulations indicate that greater surface warming trend over time occurs at higher altitudes for the mountainous regions all over the world (Gao et al., 2018; Guo et al., 2019; Mountain Research Initiative EDW Working Group, 2015; Palazzi et al., 2017; Pepin et al., 2019; Rangwala and Miller, 2012; You et al., 2020). Rangwala and Miller (2012) reviewed elevation-dependent warming (EDW) and its possible causes over four high mountain regions, i.e., the Swiss Alps, the Colorado Rocky Mountains, the Tibetan Plateau and the Tropical Andes. Their examinations found that the available observations indicate that some mountain regions show much greater warming rates at seasonal scales. The mechanisms that can produce enhanced warming rates at higher altitudes may be related to differential sensitivities of surface warming to changes in the climate drivers at different elevations, such as snow-ice cover, clouds, atmospheric water vapor, aerosols, land use, and vegetation (Rangwala and Miller, 2012; You et al., 2020).

Tibetan Plateau (TP, hereafter), known as "Third Pole" and "the roof of the world", is an ideal place to examine EDW and its mechanism (Guo et al., 2021). The warming rates (rising temperature per 10 years) over TP were found to be the most notable in winter and autumn (Liu and Chen, 2000), especially for the central and eastern Plateau (Duan and Wu, 2006), which may be partly associated with human activities, such as more anthropogenic emissions in the sub-regions (Lu et al., 2010). The absorbing aerosols (black carbon and dust) from local emissions or long-range transport heat the atmosphere in two ways. They absorb radiation and decrease the surface albedo when deposited on snow and ice (Kang et al., 2019; Lau et al., 2010; Xu et al., 2009). Ramanathan and Carmichael (2008) suggested that black carbon (BC) in the Himalayas arising from anthropogenic activities at Indo-Gangetic Plain could account for half of the local warming during the past several decades. In addition to the well-known BC, the recent work by Wu et al. (2018) suggested that the light absorption efficiency (LAE) of brown carbon (BrC, certain type of organic aerosols) in winter is 2-3 times higher than that in summer for the central Tibetan Plateau. However, the scientific knowledge on optical properties of carbonaceous aerosols (EC, BrC) is extremely limited over the Eastern TP, and *in-situ* aerosol observations at varying altitudes from the heavily polluted regions to the relatively clean Plateau were important for better understanding their light absorption of over the TP.



The previous *in-situ* observations mainly focused on the southern and northern slopes (Cong et al., 2015; Huang et al., 2007; Kang et al., 2020), while less observations were conducted at the eastern slope of the TP (ESTP). Sichuan Basin (SCB), a highly polluted region in China, is located in the east

side of TP (Zhao et al., 2018). The BrC LAE was strong inside the basin (Peng et al., 2020a), especially for the rural areas due to more biomass and coal burning impacts (Zhao et al., 2021). Our previous works indicated that aerosols from SCB are transported upslope along ESTP and reach eastern part of TP by gradient *in-situ* observations at ESTP (Yin et al., 2020). The recent paper by S. Y. Zhao et al. (2020) suggested the strongly light-absorbing BrC from biomass and coal burning inside the basin can

be transported to main part of TP by the enhanced "heat pump" due to rapid warming over the TP. The aerosols over the TP from local emissions and long-range transport from the surrounding highly polluted areas affected its weather, climate and water cycle (C. F. Zhao et al., 2020). The clouds and radiation are particularly sensitive to aerosols over pristine regions (Garrett and Zhao, 2006; Zhang et al., 2021). However, it is fuzzy that changes in light absorption and radiative forcing of carbonaceous

aerosols from the highly polluted SCB to the cleaner TP.

In this work, we investigated the changes in light absorption of carbonaceous aerosols (EC, BrC) and calculated relative radiative forcing of BrC to EC aerosols from SCB to TP in the four seasons. The sources and origins also were determined by some statistical methods and HYSPLIT back trajectory

model. Our goals are to understand EC or BrC light absorption difference between the highly polluted basins and clean TP and to reveal the corresponding mechanisms and to provide a basic data set for optimization of regional climate modeling.

## 2 Data and methods

### 2.1 Observation sites and aerosol sampling

Compared with the coarser fraction of PM, strong light-absorbing carbonaceous particles are mainly located in submicron range. Therefore, $PM_1$ (particulate matter with aerodynamic diameter smaller than 1 μm) samples were collected at six sites (Chengdu, Sanbacun, Wenchuan, Lixian, Maerkang and Hongyuan) from western SCB to east part of TP with varying elevation from 500 m to 3500 m (Figure

1). Each sampling site is selected to represent background level at local scale as completely as possible



without local emission impacts. The 1024 $PM_1$ samples in total were collected from December 21,

2018 to December 18, 2019 on a day / night pattern by aerosol sampler (LY-2034, Laoying Instrument

Co., Ltd., China) at the flow rate of 100 L min$^{-1}$. The samples were stored frozen in prebaked glass jars

until further analysis (Kawamura et al., 2010). The meteorological variables (temperature, relative

5    humidity, wind speed and direction) were downloaded by China Meteorological Data Service Center

(http://data.cma.cn/). The MODIS active fire data (https://earthdata.nasa.gov/active-fire-data) also were

used in this study.

**2.2 Chemical analysis**

10    A quarter of each filter was used to analyze water-soluble inorganic ions ($Na^+$, $NH_4^+$, $K^+$, $Ca^{2+}$, $Mg^{2+}$,

$F^-$, $Cl^-$, $SO_4^{2-}$, and $NO_3^-$), and the ions were extracted and filtered by ultrapure water and a 0.45 μm pore

syringe filter. The concentrations of the cations and anions were measured by ion chromatograph (DX-

600 & ICS-2500, Dionex, USA). The carbonaceous aerosols, i.e., Organic carbon (OC) and elemental

carbon (EC), were determined by a seven-wavelength carbon analyzer (DRI-2015, USA). The carbon

analyzer measured OC and EC concentrations using the thermal/optical reflectance (TOR) method

(Chow et al., 2007). Briefly, the OC / EC was determined by progressively heating the sub-filter. The

OC fractions were determined by heating at 120 °C (OC1), 250 °C (OC2), 450 °C (OC3) and 550 °C

(OC4) in a pure He atmosphere; subsequently, EC fractions were measured at 550 °C (EC1), 700 °C

(EC2) and 800 °C (EC3) in an oxidizing atmosphere of 2% $O_2$ and 98% He. The involved carbon is

oxidized to $CO_2$ and then reduced to $CH_4$ for detection by a flame ionization detector. The pyrolyzed

organic carbon (OPC) was monitored when the reflected laser signal returned to its initial value after

introducing $O_2$ to the analysis atmosphere. The OC was defined as the sum of OC1, OC2, OC3, OC4

and OPC while EC was defined as EC1 + EC2 + EC3 − OPC.

The coefficient of determination (COD) in conjunction with correlation coefficients (r) can be used to

characterize intra-location variability of chemical species (Zhao et al., 2021). COD is calculated by the

below equation:

$$COD_{jk} = \sqrt{\frac{1}{p}\sum_{1}^{p}\left(\frac{x_{ij}-x_{ik}}{x_{ij}+x_{ik}}\right)^2} \qquad (1)$$

where $x_{ij}$ and $x_{ik}$ are the average concentration for a chemical component $i$ and site $j$ and $k$, and $p$ is the





number of samples. The COD values of zero and approaching one mean no difference and absolute

heterogeneity between the two sites for the specific chemical component, respectively. The COD lower

than 0.2 is usually considered to represent relatively similarity of spatial pattern (Wang et al., 2018).

**2.3 Calculation of light absorption properties**

The BrC light absorption increases sharply as decreased wavelength, and thus it can be separated from

EC (Peng et al., 2020a). The light absorption induced by carbonaceous aerosols (sum of EC and BrC)

on a quartz filter was estimated by an algorithm of transmittance attenuation (*ATN*):

$$ATN_\lambda = \ln\left(\frac{FT_{\lambda,a}}{FT_{\lambda,b}}\right) \qquad (2)$$

where, $FT_{\lambda,a}$ and $FT_{\lambda,b}$ in the right hand represent filter transmittance after and before thermal analysis

for the specific wavelength ($\lambda$). Referring to the work by Chen et al. (2015), the relation of ATN with

absorption optical depth ($\tau_a$) can be given as follows,

$$\tau_{a,\lambda} = a_\lambda \times ATN_\lambda^2 + c_\lambda \times ATN_\lambda \qquad (3)$$

This study used the two coefficients ($a_\lambda$ and $c_\lambda$) reported by Chen et al. (2015). The light absorption

coefficients ($b_{abs}$) can be calculated with the equation:

$$b_{abs,\lambda} = \tau_{a,\lambda} \times \left(\frac{A}{V}\right) \qquad (4)$$

where, *A* and *V* are filter area and sampling volume, respectively. The total $b_{abs}$ can be separated into

EC and BrC by a simplified two-component model (Chen et al., 2015):

$$b_{abs,\lambda} = b_{abs,\lambda,EC} + b_{abs,\lambda,BrC} = K_1 \times \lambda^{-AAE_{EC}} + K_2 \times \lambda^{-AAE_{BrC}} \qquad (5)$$

where, $K_1$ and $K_2$ are fitting coefficients. $AAE_{EC}$ and $AAE_{BrC}$ represent EC and BrC absorption

Ångström exponent (AAE), respectively. These are wavelength independent factors. $AAE_{EC}$ was

assumed as 1 (Bond, 2001), and the other three parameters in Eq. (5) were obtained for $AAE_{BrC}$ values

between 2 and 8 with the increment of 0.1 by least-square linear regression, and the $AAE_{BrC}$ that led to

the overall best fit in terms of $R^2$ was selected as the effective BrC AAE. The mass absorption

efficiency (MAE) was obtained by the ratio of light absorption coefficients ($b_{abs,\lambda,EC}$ or $b_{abs,\lambda,BrC}$) to the

corresponding EC or OC mass concentrations (Olson et al., 2015).



The absorbed light by carbonaceous component can be estimated as follows (Huang et al., 2018):

$$\frac{I_0 - I}{I_0}(\lambda, EC) = 1 - e^{-\left(MAE_{\lambda 0, EC} \times \left[\frac{\lambda_0}{\lambda}\right]^{AAE_{EC}} \times C_{EC} \times PBLH\right)} \tag{6}$$

$$\frac{I_0 - I}{I_0}(\lambda, BrC) = 1 - e^{-\left(MAE_{\lambda 0, BrC} \times \left[\frac{\lambda_0}{\lambda}\right]^{AAE_{BrC}} \times C_{OC} \times PBLH\right)} \tag{7}$$

where, 405 nm is determined as reference wavelength $\lambda_0$, and $C_{EC}$ and $C_{OC}$ represent EC and OC

concentrations, respectively. The planetary boundary layer height (PBLH) was obtained from the

HYSPLIT model, and we assumed no vertical gradients within the PBL. The radiative forcing of BrC

relative to EC can be estimated by the below equation (Zhao et al., 2019):

$$f = \frac{\int I_0(\lambda)\left[\frac{I_0 - I}{I_0}(\lambda, BrC)\right]d\lambda}{\int I_0(\lambda)\left[\frac{I_0 - I}{I_0}(\lambda, EC)\right]d\lambda} \tag{8}$$

where $I_0(\lambda)$ is wavelength-dependent solar emission flux, which is clear sky Air Mass 1 Global

Horizontal solar irradiance (Levinson et al., 2010). The light absorption by BrC at 405 nm and 445 nm

is much stronger than that in the longer wavelength inside SCB (Zhao et al., 2021). Therefore, the

fraction ($f$) is obtained by numerical integration of the above formula in the wavelength range of 405-

980 nm and 405-445 nm for each sample, respectively.

The exponential function was selected to fit the relationships between BrC MAE and altitude (*AT*). The

equation is given as follows:

$$MAE_{\lambda, BrC} = a_\lambda \cdot e^{b \times AT} \tag{9}$$

where, $a_\lambda$ and $b$ are the fitted coefficients, and *AT* is altitude. The EC MAE can be parameterized with

altitude by replacing the subscript of *BrC* with *EC* in Eq. (9).

### 2.4 HYSPLIT backward trajectory model

HYbrid Single-Particle Lagrangian Integrated Trajectory (HYSPLIT) model developed by the National

Oceanic and Atmospheric Administration's (NOAA) is a complete system for computing simple air

parcel trajectories (Draxler et al., 2009). HYSPLIT continues to be one of the most extensively used

atmospheric transport and dispersion models. A common application is a back trajectory analysis to

determine the origin of air masses and establish source-receptor relationships. In this study, HYSPLIT



model was used to determine potential source regions of air pollutants in the four seasons at the six sites. The 96-h backward trajectories arriving at 500 m above ground level (AGL) and initializing at each hour of day were calculated with $0.25° \times 0.25°$ Global Data Assimilation System (GDAS) data from National Centers for Environmental Prediction (NCEP). The gridded back trajectory frequencies were

calculated with Openair package of Rplot.

### 2.5 PMF receptor model

EPA PMF receptor model is a mathematical approach for quantifying the contribution of sources to samples based on the composition or fingerprints of the sources. A speciated data set can be viewed as

a data matrix X of $i$ by $j$ dimensions, in which $i$ number of samples and $j$ chemical species were measured, with uncertainties $u$. The goal of PMF model is to solve the chemical mass balance between measured species concentrations and source profiles, as shown in the below Eq. (10), with number of factors $p$, the species profile $f$ of each source, and the amount of mass $g$ contributed by each factor to each sample:

$$x_{ij} = \sum_{k=1}^{p} g_{ik} f_{kj} + e_{ij} \qquad (10)$$

where $e_{ij}$ is the residual for each sample/species. In this study, the uncertainties of the chemical species concentrations were estimated by the Eq. (11):

$$Unc = \sqrt{\left(0.1 \times concentration\right)^2 + \left(0.5 \times MDL\right)^2} \qquad (11)$$

where MDL is species-specific method detection limit. The water-soluble ions and carbonaceous

aerosols in the four seasons at the six sites were used as input variables to run PMF model. The MDL of the species can refer to Cui et al. (2019).

### 3 Results and discussion

### 3.1 Light absorption of EC and BrC

The effect of carbonaceous aerosols on regional and even global climate is more uncertain due to short life than the long-lived ones, such as carbon dioxide and methane (Chung et al., 2012; Ramanathan and Carmichael, 2008). Table 1 shows seasonal mean OC and EC concentrations, light absorption coefficient and efficiency ($b_{abs}$, MAE) of EC and BrC at 405 nm and the corresponding meteorological


variables at the six sites during the campaign. The winter mean OC (EC) concentrations ranging from 5.3 (2.2) μg m$^{-3}$ at Maerkang to 18.9 (7.9) μg m$^{-3}$ at Sanbacun is about 2–6 times higher than those in the other seasons, which is mainly related to more primary emissions in winter with similar wind speeds (Table 1). The much higher OC/EC ratios at the plateau sites than that at the basin sites suggests

that more secondary OC is formed by chemical reactions over Tibetan Plateau, which can be supported by the works of Wu et al. (2018). Combined with the previous studies, the winter OC concentration is found to vary from 15.0 to 20.1 μg m$^{-3}$, while EC is between 4.3 and 4.7 μg m$^{-3}$ at urban areas inside SCB, which is significantly lower than that at Indo-Gangetic Plain (Table S1). However, OC and EC concentrations at eastern TP are much more abundant than that at western and southern TP sites due to

more dense population and industry (Table S1). Briefly, carbonaceous aerosol pollution is much more severe inside the basin than that over TP, indicating that the large amount of air pollutants is trapped inside the basin due to calm and stable air.

Figure 2 compares spectral total and separated b$_{abs}$ from EC and BrC in spring and winter at the six

sites along the eastern slope of Tibetan Plateau. The measured (green hollow points) and calculated b$_{abs}$ (yellow dash lines) for total carbon (TC, sum of EC and BrC) is comparable, and the difference is within 5%. For Sanbacun, a rural site inside the basin, the b$_{abs}$ is much higher than the other sites, especially for the shorter wavelength due to more BrC emissions from coal and biomass burning for cooking and heating at rural areas inside SCB (Zhao et al., 2021). The absorption due to EC decreases

with altitude primarily because EC concentration decreases (see Table 1). This may be partly due to stable air inside the deep basin (Feng et al., 2020), but that would also apply to BrC in so far as EC and BrC share sources, and vertical mixing is primarily due to fair weather convection rather than deep convective storms (Zhang et al., 2017). However, the light absorption by BrC is not monotonically changed as altitude due to more complicated sources and origins of BrC. The b$_{abs}$ at 405 nm of BrC

accounting for TC increases from 20% for Chengdu to ~ 50% for Hongyuan, while the proportion significantly reduces as increased wavelength (Figure S1), suggesting that light absorption by BrC is much stronger at high altitudes than that at lowlands.

Compared with b$_{abs}$, MAE can better reflect light absorption efficiency of carbonaceous aerosols. The

winter mean EC MAE is 6.0±1.0 m$^2$ g$^{-1}$ among all sites, which is within the range of 3.9–11.9 m$^2$ g$^{-1}$





over TP and the surrounding basin regions (Tables 1 and S1). Except our result in the rural site, the

winter mean BrC MAE of 0.7–0.8 $m^2$ $g^{-1}$ inside the SCB is about half of that at Indo-Gangetic Plain

(IGP) probably due to more BrC emissions, and PM size distribution and composition difference

between SCB and IGP (Choudhary et al., 2018). Figures 3 and S2 show box plots of spectral BrC and

EC MAE in the four seasons from the basin to plateau sites extending elevation from 0.5 to 3.5 km.

Different from EC, BrC MAE at 405 nm over TP is 2–3 times higher than that inside SCB with

strongly elevation-dependent light-absorbing, and the only clear dependence is in winter. Wu et al.

(2018) found that winter BrC MAE is 4.5 $m^2$ $g^{-1}$ for a pristine environment over TP (Nam Co, 4730 m

asl), which is significantly higher than that at Hongyuan (3500 asl) for our study. The winter average

OC/EC ratio of 14.1 at Nam Co is largely higher than that at our sampling sites. Therefore, the clearly

increased BrC MAE with altitude in winter may be related to BrC composition seasonally, while EC

MAE decreases with altitude in winter may be due to the difference in source composition and aging

aerosols inside the deep basin (Liu et al., 2020). The mechanism will be discussed in the following

sections.

Figure 4 shows $MAE_{BrC}$ and $MAE_{EC}$ variations as altitude in spring and winter during the campaign.

The contrasting MAE variations as altitude between BrC and EC in winter are more significant than

those in spring. The better relationships in winter may be because more urban and aged sources are

trapped inside the deep basin in response to strong temperature inversion in winter (Feng et al., 2020).

The relation of $MAE_{BrC}$ or $MAE_{EC}$ at 405 nm with altitude can be parameterized with exponential

function (Eq. 9). The spring and winter $MAE_{BrC}$ can be parameterized with altitude ($AT$) as follows:

$$MAE_{405,BrC,spr} = 1.33 \cdot e^{0.18 \cdot AT} \tag{12}$$

$$MAE_{405,BrC,win} = 0.82 \cdot e^{0.33 \cdot AT} \tag{13}$$

Similarly, the winter $MAE_{EC}$ can be parameterized by altitude ($AT$) as follows:

$$MAE_{405,EC,win} = 11.35 \cdot e^{-0.18 \cdot AT} \tag{14}.$$

### 3.2 Sources impacting on light absorption of EC and BrC

OC/EC ratio can be used to roughly identify sources of carbonaceous aerosols, and the ratio of aerosols

from fossil combustion is generally lower than that of biomass burning (Bond et al., 2004). Figure S3





shows the relationship between OC and EC concentrations inside Sichuan Basin and that over Tibetan

Plateau during the campaign, and OC/EC ratio was obtained by fitting the relationships with univariate

linear regression. The significantly simultaneous change between OC and EC ($R^2=0.80$ for SCB, and

$R^2=0.75$ for TP) indicated that the sources may be similar. OC/EC ratio of 2.14 for western SCB and

2.06 for eastern TP is significant lower than that at Nam Co (13.8–14.1, Wu et al., 2018) representing a

pristine environment over central TP (Cong et al., 2009), while the ratios are much higher than that at

Lhasa, the largest city over TP (1.46, Li et al., 2016). The ratio of OC to EC for our study is slightly

lower than that at urban areas in eastern China and Helsinki, Finland (Han et al., 2014; Viidanoja et al.,

2002), indicating carbonaceous aerosols at western SCB and eastern TP may be significantly affected

by primary sources.

Besides primary sources, secondary formation largely contributed to OC aerosols, and thus secondary

organic carbon (SOC) was considered with EC-tracer method (Turpin and Lim, 2001). To better

understand light absorption of OC from primary emissions and secondary formation, Figures S4 and 5

show sample-to-sample and mean $MAE_{BrC}$ variations as SOC and POC concentrations for each site in

spring and winter during the campaign, respectively. The light absorption efficiency of BrC

significantly declines as the increased OC composition with the better relationships for POC at each

site (Figure S4). The mean winter $MAE_{BrC}$ decreased by about 70% as POC increases from 3.0 μg m$^{-3}$

at Hongyuan to higher than 20 μg m$^{-3}$ at Chengdu (Figure 5). SOC accounting for OC significantly

increases from western SCB to eastern TP, and it is more than 50% at Maerkang and Hongyuan due to

relatively less primary sources over TP. The large $MAE_{BrC}$ increment as SOC/POC ratio in winter

indicates that the more SOC and the less POC was favorable for BrC light absorption enhancement

(Figure 5). Therefore, the strong elevation-dependent $MAE_{BrC}$ in winter (Figure 4) may be induced by

SOC/POC ratio variations from the western SCB to TP.

The EC light absorption efficiency largely reduced as EC concentrations increase for each site in spring

and winter during the campaign (Figure S5). However, the winter mean $MAE_{EC}$ inside the basin with

high EC values was much larger than that over the TP with low EC values, while for similar EC

concentrations among the plateau sites, $MAE_{EC}$ at Wenchuan is about 2 times higher than that at

Hongyuan with strong dependence on elevation. Therefore, winter aerosol aging inside the deep basin





and larger changes in sources from the western SCB to eastern TP may induce enhanced light absorption at the lowlands than over TP. The increased in different degrees $MAE_{EC}$ as the ratios of water soluble ions ($K^+$, $Cl^-$, $SO_4^{2-}$, and $NO_3^-$) to EC concentrations suggests that EC light absorption is certainly impacted by many anthropogenic sources at the six sites (Figure S6). To further find key

sources impacting EC MAE, we check the spring and winter mean $MAE_{EC}$ variations as concentrations of chemical species from anthropogenic sources at the six sites (Figure 6). Compared with $MAE_{EC}$ in spring, the winter $MAE_{EC}$ is the lower due to high EC concentrations and more sensitive to the chemical species from anthropogenic emissions. Furthermore, $NO_3^-$ difference among the sites (Figure 6a) is much larger than $K^+$, $Cl^-$ and $SO_4^{2-}$ due to many fossil fuel combustion at Chengyu City Clusters

inside the basin. The spatial heterogeneity in ($NO_3^-$+ $SO_4^{2-}$) / ($K^+$+ $Cl^-$+ $NO_3^-$+ $SO_4^{2-}$) ratio in winter is more significant than that in spring, and winter $MAE_{EC}$ obviously increases as the ratio from TP to the basin sites. Therefore, the emissions from fossil fuel may be a key factor influencing $MAE_{EC}$ in winter.

The above paragraphs separately showed light absorption efficiency of BrC and EC and their variations

as chemical species, and the change in radiative forcing of BrC relative to EC ($f$) from Chengdu inside the western SCB to Hongyuan over eastern TP is showed in Figure 7a to reveal the mechanism. The altitude ($AT$) increased by 3 km, while the median radiative forcing of BrC relative to EC increases from about 0.10 inside the basin to 0.42 over eastern TP. The relationship between $f$ and altitude can be parameterized as the below equation:

$$f = 0.077 \cdot e^{0.480 \cdot AT} \qquad\qquad (15)$$

Some studies found that the direct radiative forcing of BrC / (BrC+EC) increases with altitude simply due to the fact that the concentration of BrC decreases more slowly with altitude than does EC (Liu et al., 2014, 2015; Zeng et al., 2020; Zhang et al., 2017). Therefore, we also checked the variations of median OC/EC ratio during the campaign from the basin to plateau sites (Figure 7b). The OC/EC ratio

changes within the range of 2–4, and the $75^{th}$ percentiles of the ratio increase more significantly than the median values from the basin to plateau sites. Therefore, the increased radiative forcing of BrC relative to EC from western SCB to eastern TP may be closely related to more secondary formation and less primary emissions over TP than SCB (also see Figure 5c).



PMF receptor model is widely used to apportion the sources influencing air pollutants at a specific site based on the fingerprints of the sources, for example $K^+$ and $Cl^-$ were usually used as tracer for biomass burning (BB) and coal combustion (CC), respectively (Tao et al., 2016). PMF analysis is conducted in this study for each season, and motor vehicles, biomass and coal burning, dust, sea salt and secondary

formation are found to be the main sources at the six sites. Figure 8 shows mass concentrations of species for each source at each site apportioned by PMF model in winter during the campaign. The source apportionment for the other seasons are illustrated in Figures S7–S9. The winter $NO_3^-$ concentrations for secondary nitrate decrease from 3.44 µg m$^{-3}$ at Sanbacun to 0.07 µg m$^{-3}$ at Maerkang with high concentration of $NH_4^+$ ion, which is more heterogeneous than that in summer and fall. The

chemical species ($K^+$, $Cl^-$) from biomass burning and coal combustion decline from the basin to plateau sites, but the decline ranges in warm seasons (summer and fall) are more significant than those in the cold seasons (spring and winter) due to usage of more fuel for heating over the TP. For the sea salt source, $Na^+$ values are almost consistent among the sites in cold seasons, while those at Maerkang (0.21 µg m$^{-3}$ in summer, and 0.31 µg m$^{-3}$ in fall) and Hongyuan (0.18 µg m$^{-3}$ in summer, and 0.29 µg m$^{-3}$ in

fall) are significantly lower than those at the other sites with the lower altitude in warm seasons. The relatively high $SO_4^{2-}$ concentrations while the low contribution of $Cl^-$ for sea slat source is because that $NaCl$ is converted to $Na_2SO_4$ by the reaction with gaseous $H_2SO_4$ and depletes $Cl^-$, which was found in the previous studies (Eleftheriadis et al., 2014; Manousakas et al., 2017).

**3.3 Regional and long-range transport impacts**

The similarities of major chemical species between two sites should represent regional air pollution, while the differences should reflect local sources impacts. A comparison between basin (Chengdu, Sanbacun) and plateau sites (Wenchuan, Lixian, Maerkang, Hongyuan) about the mean mass concentrations of water-soluble ions and carbonaceous species in the four seasons is showed in Figures

9 and S10–S12. The numerical ranges between the two axes of each subplot are set to be equal to more clearly see spatial heterogeneity of the chemical species at the region. The both COD and correlation coefficients can be used to better understand intra-location variability (Wilson et al., 2005). The COD is between 0 and 1 (see Eq. 1), and the smaller value indicates the more uniform particle concentrations. The moderate differences are observed for the chemical species from anthropogenic

sources ($NH_4^+$, $K^+$, $SO_4^{2-}$, $NO_3^-$, $F^-$, $Cl^-$, OC, EC) in the four seasons due to relative high COD (0.22–



0.75) during the campaign. The differences indicated that there are limited similarities between basin and plateau sites, and the discrepancies were in major anthropogenic sources. The spatial heterogeneity for $K^+$ and $NO_3^-$ between basin and plateau sites is more obvious than the other species in the four seasons with the largest COD values, which is mainly related to more biomass burning and vehicle

emissions inside Sichuan Basin (Zhao et al., 2021). Furthermore, the COD values for $K^+$, $NO_3^-$, and EC in winter are significantly lower than those in the other seasons due to increased biomass burning and coal combustion for heating in winter over Tibetan Plateau. Unlike COD, high correlation coefficient for the specific chemical component does not necessarily indicate uniformity, which may suggest source similarity between sites. The correlation between basin and plateau sites largely depends on

season (Figures 9 and S10–S12). The significant correlations for $NH_4^+$, $K^+$, $SO_4^{2-}$, $NO_3^-$, OC, EC in the spring and winter infer that basin and plateau sites share similar sources for the species, while weak correlations for $NO_3^-$, OC, EC in summer and fall indicate that dissimilar sources impacts inside Sichuan Basin and over Tibetan Plateau.

Compared with the species from anthropogenic sources, the $Na^+$, $Mg^{2+}$, and $Ca^{2+}$ concentrations are more comparable between basin and plateau sites and thus COD values are the lowest among the species in the four seasons. Furthermore, changes in $Na^+$ values are more synchronous than $Mg^{2+}$ and $Ca^{2+}$ between the basin and plateau sites in summer and fall. The relative low COD values and high correlation coefficients for $Na^+$ and $Cl^-$ concentrations in summer and fall suggests that $Na^+$ ion at the

whole region may be affected by long-range transport from the surrounding seas (Figures S10-S11). $Na^+$ concentration also is found to be high in salt-rich dust from saline soils (Quick and Chadwick, 2011). Dust events frequently occurred in spring and winter over Tibetan Plateau and Northwest China where saline and alkaline land and dried salt-lakes located (Jiang et al., 2021; Zhang et al., 2009), and thus the weak correlations for $Na^+$, $Mg^{2+}$, and $Ca^{2+}$ values in spring and winter between basin and

plateau sites may suggest local and regional dust plume impacts.

South Asia is found to be one of the most polluted regions, and particulate matter (PM) can be transported and deposited in southern Tibetan Plateau to accelerate ice and snow melting by reducing surface albedo (Zhang et al., 2021). MODIS active fire data suggests that biomass burning is mainly

located in South Asia around our study regions, which is more abundant in cold (spring and winter)





than warm seasons (summer and fall) during our campaign (Figure 10). The PM mass concentrations in conjunction with wind speed and direction can be used to identify the local PM origins, and thus Figure 11 shows $K^+$ pollution rose in the four seasons at the six sites. The back trajectory calculation can give regional PM origins from long-range transport, and Figures 12 and S13–S15 illustrate the gridded back

trajectory frequencies in the four seasons. As a tracer of biomass burning, $K^+$ stratification in warm seasons is more obvious than that in cold seasons, which infers that there are more biomass burning plumes over Tibetan Plateau in spring and winter. The changes in wind direction are not obvious from Sichuan Basin to Tibetan Plateau in warm seasons. However, the predominant wind direction is northwest–southeast in cold seasons for the basin sites, while that mainly focuses on southwest for the

plateau sites (Figure 11). The highest frequency of back trajectories also are in Southwest of the sampling sites in winter (Figure 12). Therefore, the biomass burning emissions originated from South Asia are transported to eastern Tibetan Plateau by highly frequent southwest winds, and thus induce high concentrations of $K^+$ ion in spring and winter.

The $K^+$ concentrations less depend on wind direction in warm seasons (Figure 11), and there are many active fire in western Sichuan Basin as compared with the cold seasons (Figure 10), and thus $K^+$ ion may be mainly affected by local biomass burning in summer and fall. Although less active fire is detected in winter (Figure 9d), the local biomass and coal combustion for winter heating was found to largely affect air quality inside the basin (Zhao et al., 2021). As one of the cloudiest areas in China (Jin

et al., 2009), the Sichuan Basin has a climatologically high cloud cover fraction of more than 80% (Qian et al., 2007), and especially in winter (Hu et al., 2021). Therefore, the active fire failed to be detected by MODIS possibly due to cloud impacts in winter.

**4 Summary and conclusions**

Tibetan Plateau (TP) is surrounded by the three highly polluted regions, i.e., Indo-Gangetic Plain (IGP), Taklimakan and Gobi Deserts (TGDs) and Sichuan Basin (SCB). However, the previous studies mainly focused on the south (IGP) and north slopes (TGDs), and thus the first *in-situ* aerosol measurements were conducted at eastern slope of Tibetan Plateau (ESTP) to study the elevation-dependent light absorption of carbonaceous aerosols from the highly polluted SCB to the pristine TP.

The sources and origins also were determined by PMF and HYSPLIT models for the six sites





(Chengdu, Sanbacun, Wenchuan, Lixian, Maerkang, and Hongyuan) extending elevation from 500 m to 3500 m. Some novel findings were obtained by this *in-situ* observations.

The EC and BrC light absorption was separated by the simple two-component model. The BrC light

absorption coefficients at 405 nm accounting for total carbon (TC, sum of EC and BrC) are found to be increased from ~ 20% inside the SCB to ~ 50% over the TP. The BrC mass absorption efficiency (MAE) over eastern TP is 2–3 times higher than that inside the SCB with strongly elevation-dependent light-absorbing. The most significant elevation-dependent MAE of BrC in winter is closely related to the high ratio of secondary to primary organic carbon (OC), i.e., more OC from secondary

transformation than primary emissions at high altitudes. Different from BrC, winter EC MAE declines from the highly polluted SCB to clean TP, which is due to source difference between the two regions. More urban sources (vehicles, industries, etc.) are trapped inside the deep SCB due to poor dispersion and frequent temperature inversion in cold seasons. The high primary emissions and weak dispersion conditions are favorable for full mixing and aerosol aging to enhance absorption inside the basin. The

median radiative forcing of BrC relative to EC increased from about 0.10 inside the basin to 0.42 over eastern TP, which is associated with OC/EC ratio. Therefore, the enhanced radiative forcing of BrC relative to EC is because the concentration of OC decreases more slowly with altitude than does EC. South Asia is determined as the main origins of PM pollutants at the study region from western SCB to eastern TP, which is significantly dependent on seasons.

The first aerosol field experiment was conducted at the specific study region, but only six sampling sites were used from the deep SCB to eastern TP in this study. The more sites will be established to better understand the chemical composition and aerosol light properties and sources and origin impacts at the study region. The light absorption coefficients and efficiencies of BrC failed to be separated from

that of TC in summer and fall at Maerkang and Hongyuan due to instrument failure, which limited to reveal the elevation-dependent light absorption. Furthermore, replacing BrC, OC mass concentration was used to estimate BrC MAE, which may have large uncertainty, and thus these are expected to be corrected in the future study.

*Data availability.* Raw data sets (Zhao et al., 2022, DOI: 10.5281/zenodo.6474199) used in this manuscript were



available at https://zenodo.org/record/6474199#.YmCn_YtByUk.

*Author contributions.* Suping Zhao and Ye Yu designed the study. Suping Zhao analyzed the data with help from

Ye Yu, Jinbei Chen and Shichang Kang. Daiying Yin and Longxiang Dong collected and analyzed data during the

campaign. Shaofeng Qi conducted the field experiment.

*Competing interests.* The authors declare that they have no conflict of interest.

*Financial support.* This work was supported by the National Natural Science Foundation of China (42075185;

41605103), Youth Innovation Promotion Association, CAS (Y2021111), and Gansu Science and Technology

Program key projects (20JR10RA037 and 18JR2RA005).

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

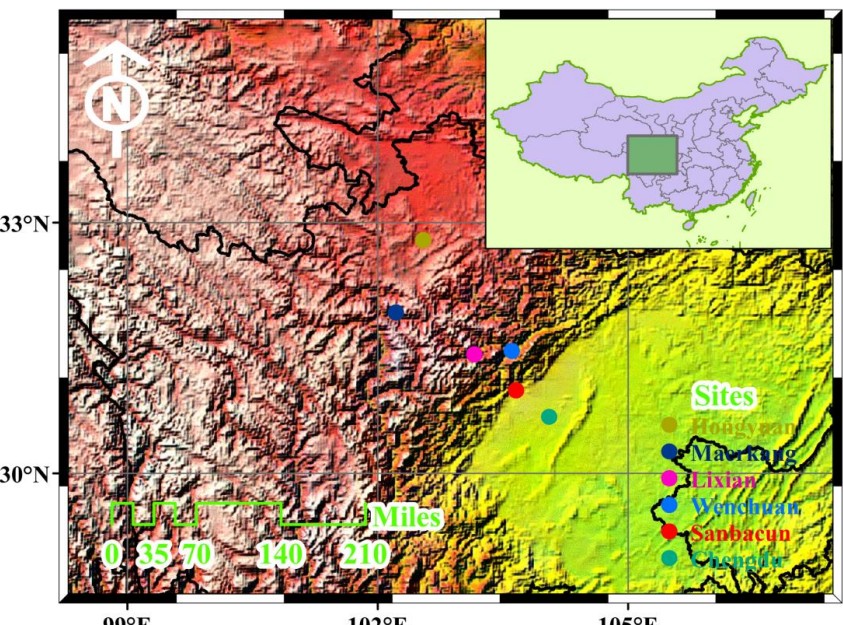

**Figure 1: Geographic location of the six** *in-situ* **observation sites (Chengdu, Sanbacun,**

**Wenchuan, Lixian, Maerkang, and Hongyuan) along the eastern slope of Tibetan Plateau. The**

**map is a pure reproduction of Google Maps with added a marks for our study locations.**

5     **Copyright © Google Maps.**



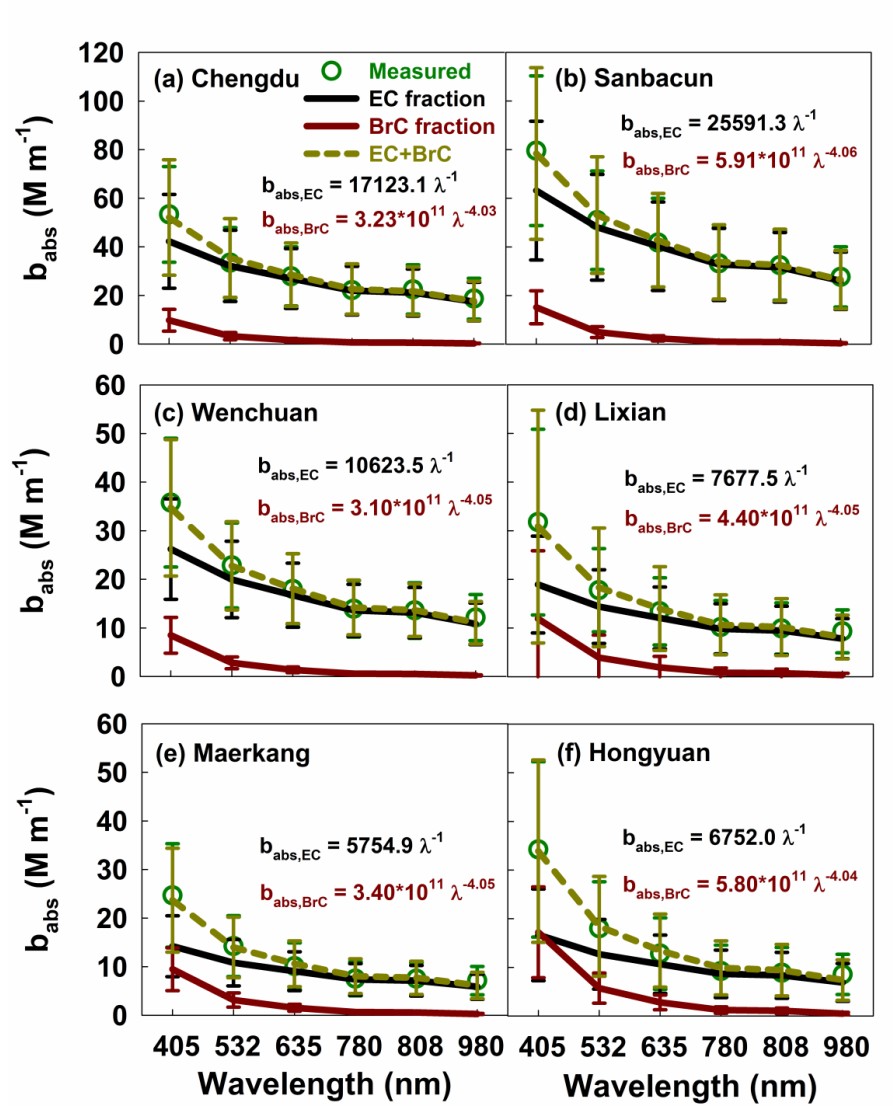

**Figure 2: Spectral light absorption coefficients ($b_{abs}$) by EC and BrC in spring and winter at the six sites along the eastern slope of Tibetan Plateau. The subplots depict the decomposition of total light absorption by EC and BrC with the model given in Eq. 4. Error bars represent uncertainties derived from replicate analyses and lower quantifiable limits.**

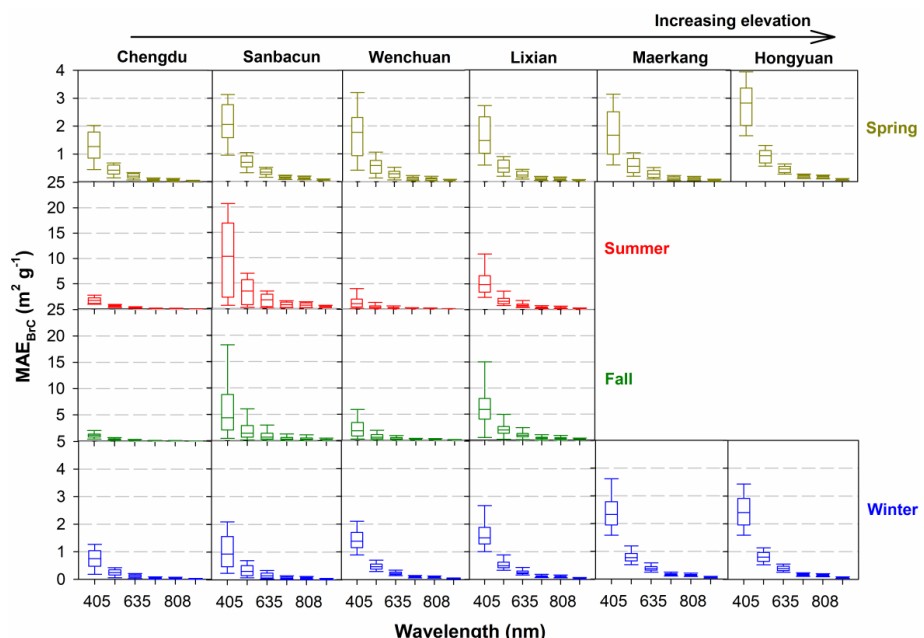

**Figure 3: Box plots of spectral BrC mass absorption efficiency (MAE$_{BrC}$) in each season from Chengdu inside SCB to Hongyuan over the TP covering elevation from 500 m to 3500 m. The lines inside the box denote the median values; the two whiskers and the top and bottom of the box denote the 5th and 95th and the 75th and 25th percentiles.**

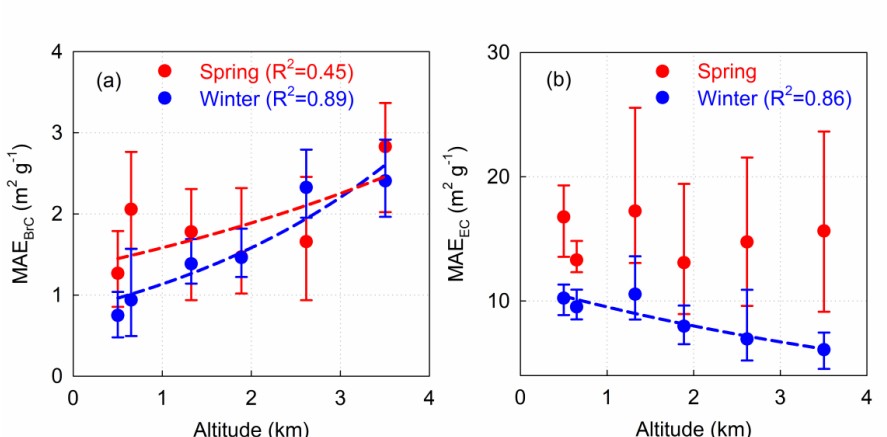

**Figure 4: Variations of (a) MAE$_{BrC}$ and (b) MAE$_{EC}$ at 405 nm as altitude in spring and winter during the campaign. The solid dots denote the median values; the two whiskers of the dots denote the 25th and 75th percentiles. The relationships were fitted by exponential function, and passed the significance level of 0.01. The coefficients of determination (R$^2$) also were given in each subplot.**



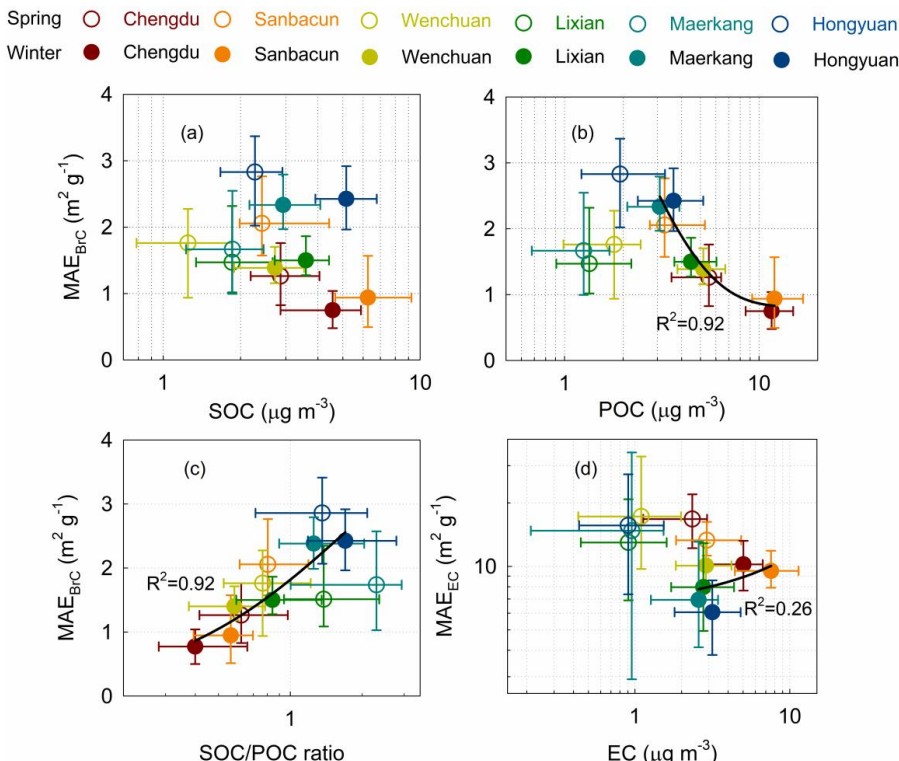

**Figure 5: Variations of spring and winter mean MAE$_{BrC}$ as (a) SOC, (b) POC, (c) SOC/POC ratio and (d) MAE$_{EC}$ as EC concentrations at the six sites. The hollow and solid dots denote the median values in spring and winter; the four whiskers of the dots denote the 25th and 75th percentiles of the corresponding two variables. The horizontal axis in each subplot is showed on a logarithmic scale to more clearly see the details.**

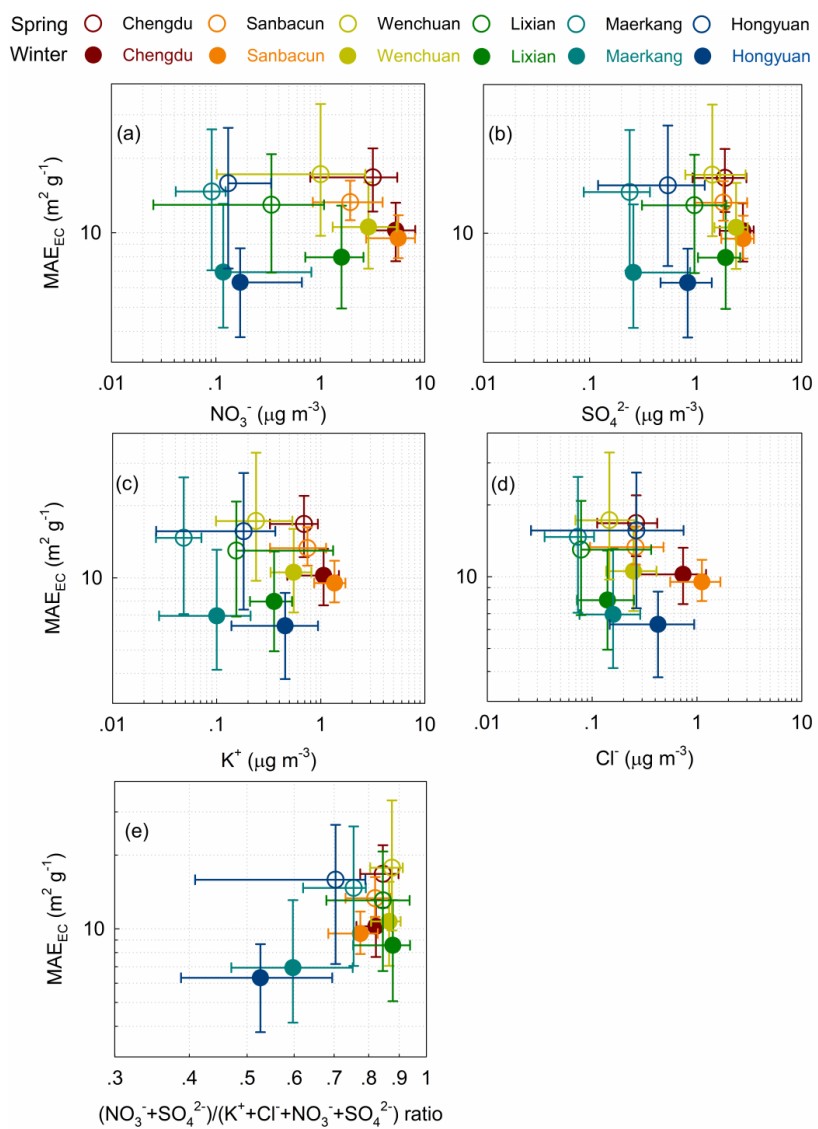

**Figure 6: Variations of spring and winter MAE$_{EC}$ as (a) NO$_3^-$, (b) SO$_4^{2-}$, (c) K$^+$, and (d) Cl$^-$**

**concentrations, and (e) (NO$_3^-$+ SO$_4^{2-}$) / (K$^+$+ Cl$^-$+ NO$_3^-$+ SO$_4^{2-}$) ratio at the six sites. The hollow**

5 **and solid dots denote the median values in spring and winter; the four whiskers of the dots**

**denote the 25th and 75th percentiles of the corresponding two variables. The axes in each subplot**

**are showed on a logarithmic scale to more clearly see the details.**

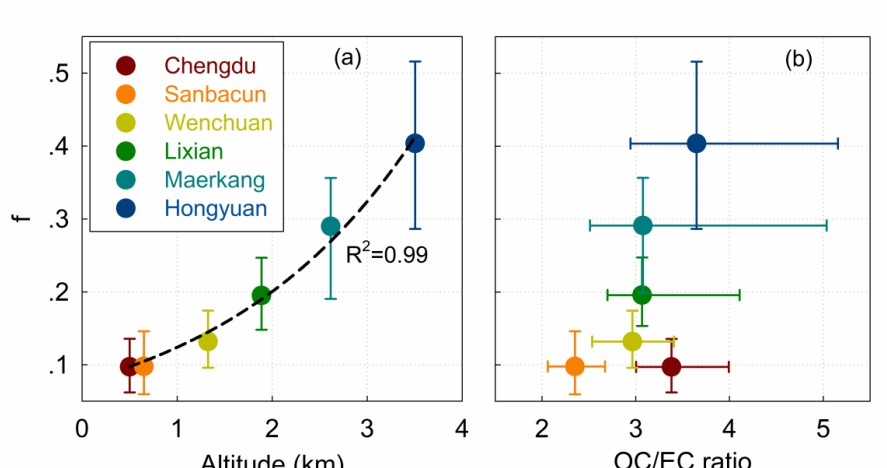

**Figure 7: Variation of radiative forcing of BrC relative to EC (f, see Eq. 7) as (a) altitude and (b) OC/EC ratio for each site. The solid dots denote the median values; the two whiskers of the dots denote the 25th and 75th percentiles of the variables. The relationship between f and altitude was fitted by exponential growth function, and passed the significance level of 0.01.**

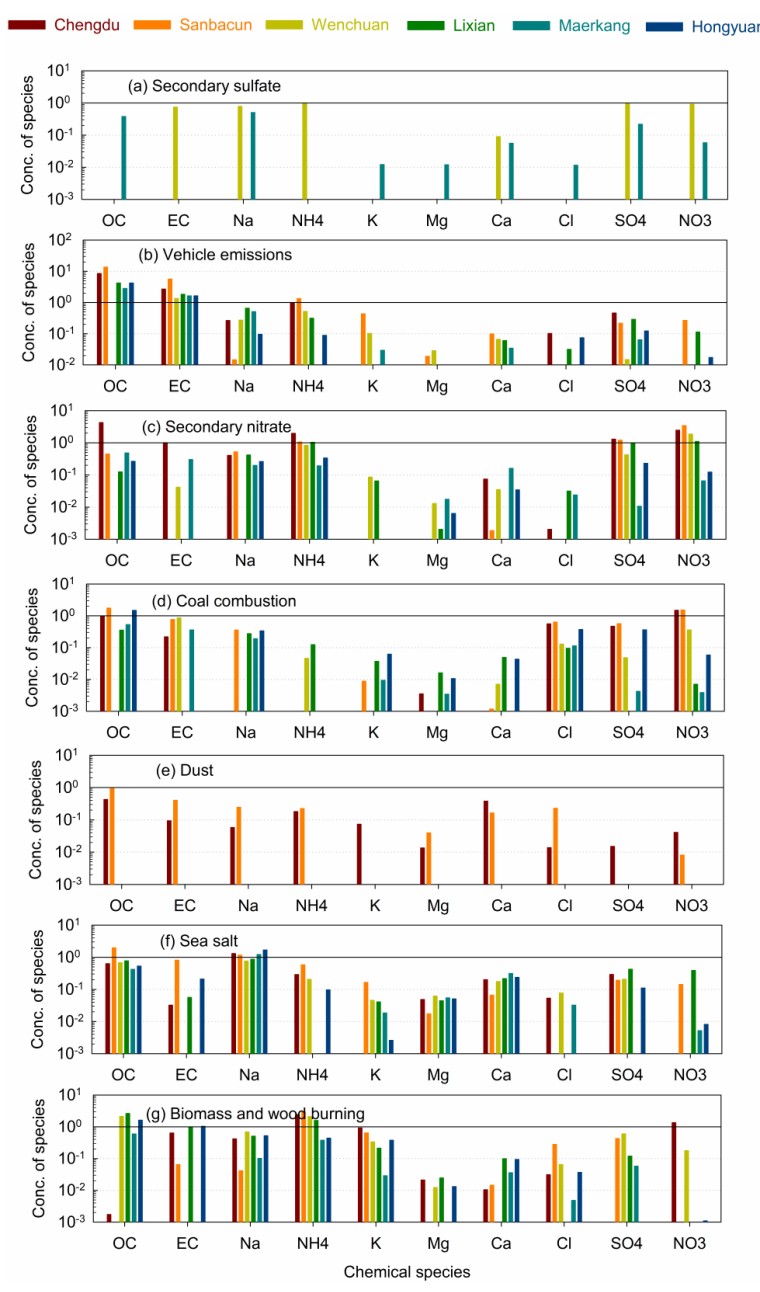

**Figure 8: Mass concentrations of species for each source at each site apportioned by PMF model in winter during the campaign. The vertical axes are showed on logarithmic scale to better distinguish the concentrations of chemical species among the sampling sites.**



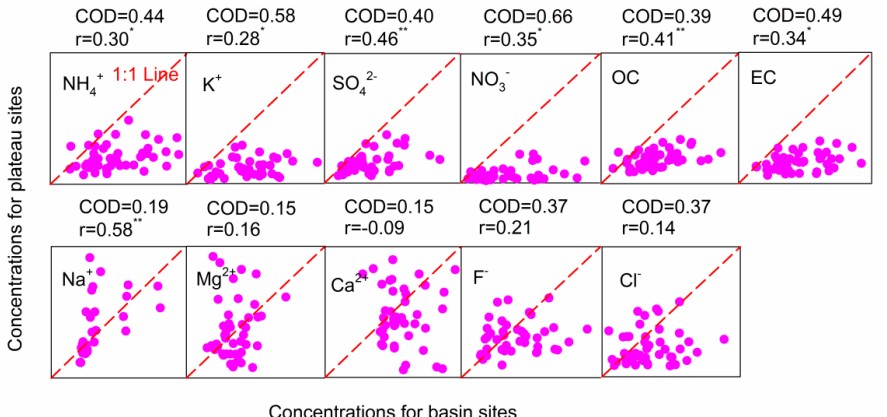

**Figure 9: Relationships of spring PM₁ chemical components concentrations between basin (horizontal axes, including Chengdu and Sanbacun) and plateau sites (vertical axes, including Wenchuan, Lixian, Maerkang and Hongyuan). The correlation coefficients (r) with an asterisk and two asterisk superscripts pass the significance level of 0.05 and 0.01, respectively.**

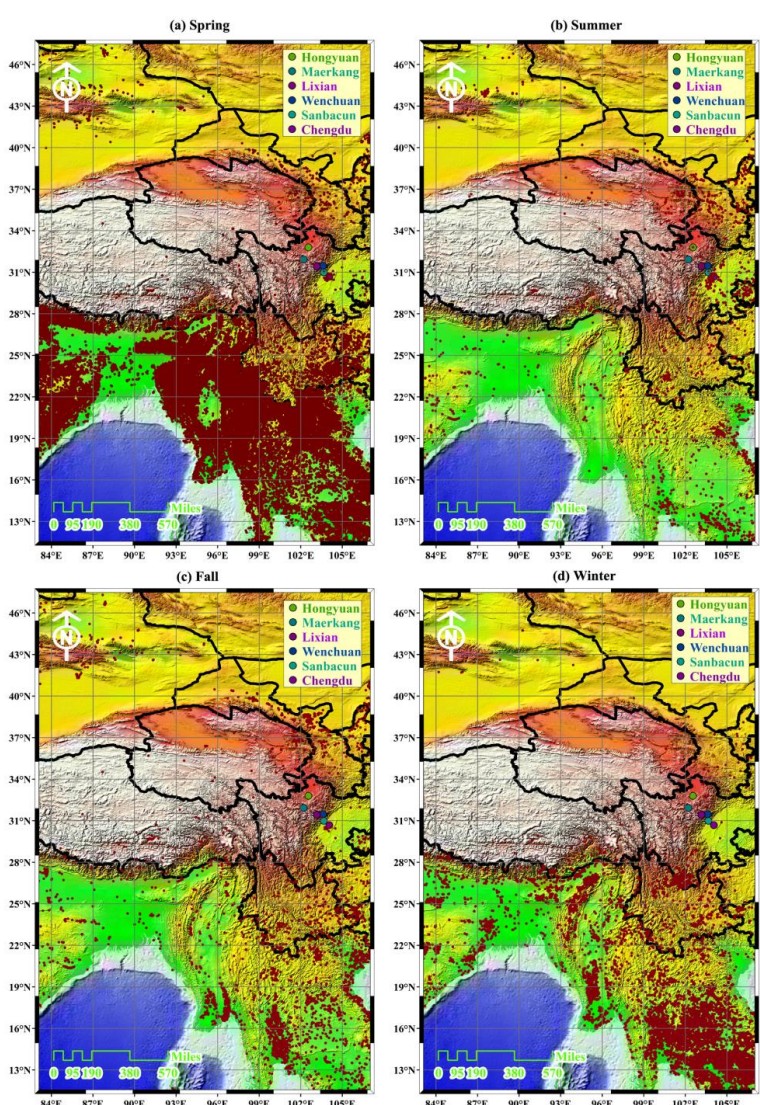

**Figure 10: MODIS active fire locations in Southeast Asia in the four seasons. The six sampling sites along the eastern slope of Tibetan Plateau also were showed in each subplot. The map is a pure reproduction of Google Maps with added the active fire data. Copyright © Google Maps.**



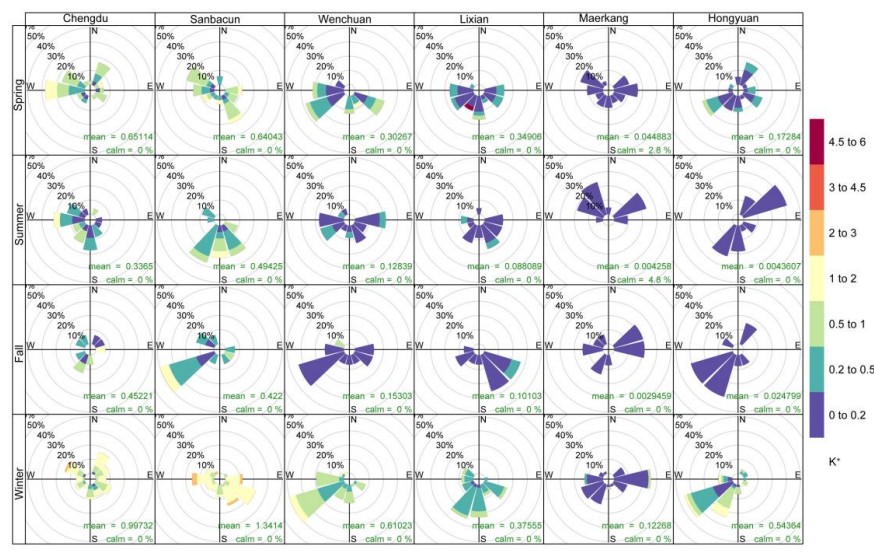

**Figure 11: K⁺ pollution rose in the four seasons at the six sites along the eastern slope of Tibetan**

**Plateau. Mean K⁺ concentrations and calm frequencies also were given in each subplot.**

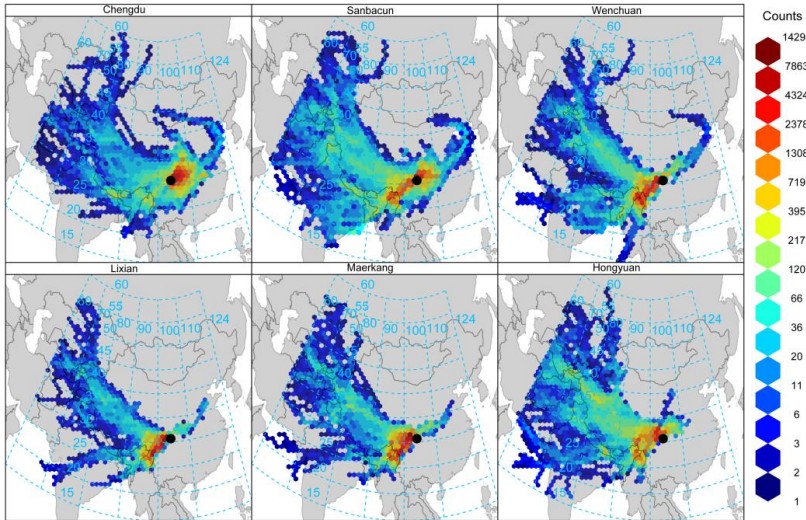

5   **Figure 12: Gridded back trajectory frequencies with hexagonal binning in winter at the six sites**

**from west Sichuan Basin to Tibetan Plateau. The map is a pure reproduction of Google Maps**

**with added the trajectory frequencies. Copyright © Google Maps.**



**Table 1 Seasonal mean values (mean ± std.) of OC and EC concentrations, light absorption coefficient (b$_{abs}$), mass absorption efficiency (MAE) and meteorological variables (wind speed (WS), temperature (Tem.), relative humidity (RH)) for the six sites during the measurement campaign. There is no b$_{abs}$ or MAE reported for MEK and HY in summer and fall as the used DRI instrument does not work at the 2 wavelength of 405 nm and 445 nm when the samples are measured, and separation of EC and BrC cannot be conducted by Eq. (5).**

| Season | Sites | OC | EC | b$_{abs}$ (M m$^{-1}$) | | MAE (m$^2$ g$^{-1}$) | | WS | Tem. | RH |
|--------|-------|-----|-----|-----|-----|-----|-----|-----|-----|-----|
| | | (µg m$^{-3}$) | (µg m$^{-3}$) | BrC, 405 | EC, 405 | BrC, 405 | EC, 405 | (m s$^{-1}$) | (°C) | (%) |
| Spring | CD | 7.9±3.7 | 2.2±1.2 | 8.5±2.8 | 32.4±12.7 | 1.3±0.6 | 17.1±4.8 | 1.6±0.7 | 17.5±4.3 | 80.3±19.9 |
| | SBC | 6.7±3.0 | 3.3±1.5 | 13.7±5.6 | 44.2±16.9 | 2.1±0.9 | 13.7±2.5 | 1.4±0.6 | 16.9±4.1 | 77.6±15.9 |
| | WC | 3.2±1.6 | 1.2±0.8 | 4.8±2.3 | 20.2±9.0 | 1.6±0.9 | 21.5±11.6 | 2.4±1.0 | 15.1±4.4 | 65.1±17.4 |
| | LX | 3.5±1.4 | 1.0±0.6 | 5.4±2.5 | 11.8±5.5 | 1.7±0.8 | 13.8±6.9 | 1.6±0.5 | 13.3±5.3 | 61.5±20.4 |
| | MEK | 3.0±1.7 | 0.8±0.6 | 4.8±2.7 | 10.2±3.6 | 1.9±1.2 | 16.6±9.4 | 1.1±0.6 | 10.6±5.5 | 62.0±26.5 |
| | HY | 4.1±1.6 | 0.9±0.6 | 11.5±4.9 | 12.9±6.2 | 2.8±0.9 | 17.3±9.8 | 2.4±1.0 | 2.4±3.6 | 70.0±16.6 |
| Summer | CD | 5.4±1.2 | 1.9±0.5 | 9.0±2.7 | 29.2±6.9 | 1.8±0.6 | 16.4±4.5 | 1.3±0.4 | 25.2±2.9 | 84.6±18.8 |
| | SBC | 2.9±1.2 | 1.5±0.7 | 21.8±15.0 | 32.6±7.9 | 10.1±7.1 | 29.8±6.5 | 1.1±0.4 | 24.1±3.0 | 82.7±13.9 |
| | WC | 2.2±0.8 | 1.0±0.5 | 2.2±1.5 | 18.9±5.5 | 1.4±1.3 | 23.5±9.5 | 1.7±0.7 | 23.1±3.2 | 64.5±16.5 |
| | LX | 2.7±0.9 | 0.8±0.5 | 13.3±5.0 | 9.5±2.7 | 5.4±2.5 | 16.4±11.3 | 1.4±0.5 | 20.9±4.0 | 65.2±18.0 |
| | MEK | 2.7±1.5 | 0.7±0.6 | — | — | — | — | 1.0±0.4 | 16.6±4.3 | 73.3±22.6 |
| | HY | 3.0±1.2 | 0.8±0.6 | — | — | — | — | 1.8±0.6 | 10.1±3.3 | 77.8±11.6 |
| Fall | CD | 4.7±1.3 | 2.3±1.0 | 5.3±2.5 | 40.6±16.6 | 1.1±0.5 | 18.3±4.0 | 1.1±0.4 | 15.6±4.9 | 88.4±10.8 |
| | SBC | 5.3±3.4 | 3.0±1.8 | 22.0±13.7 | 50.8±11.2 | 6.0±5.6 | 24.3±9.3 | 0.9±0.2 | 14.9±4.4 | 89.9±11.6 |
| | WC | 1.6±0.8 | 0.8±0.5 | 3.0±2.0 | 18.2±7.3 | 2.3±1.8 | 27.3±13.9 | 1.7±0.6 | 14.1±5.4 | 72.7±10.0 |
| | LX | 2.4±1.0 | 0.9±0.5 | 12.7±6.6 | 10.5±3.4 | 6.5±3.8 | 14.7±10.1 | 1.3±0.3 | 11.9±5.6 | 76.8±11.3 |
| | MEK | 2.3±1.2 | 0.9±0.6 | — | — | — | — | 0.9±0.4 | 8.8±5.5 | 78.4±17.0 |
| | HY | 3.4±2.2 | 1.3±1.1 | — | — | — | — | 1.9±0.7 | 0.7±5.6 | 73.9±11.0 |
| Winter | CD | 15.0±5.9 | 4.7±2.0 | 10.5±4.6 | 47.6±20.1 | 0.8±0.5 | 10.4±2.8 | 1.2±0.4 | 6.6±2.7 | 78.9±16.9 |
| | SBC | 18.9±7.6 | 7.9±3.4 | 17.1±10.2 | 74.7±27.9 | 1.2±1.0 | 9.9±2.0 | 1.0±0.3 | 5.8±2.7 | 79.2±15.0 |
| | WC | 8.2±3.1 | 2.8±1.3 | 11.2±3.2 | 29.7±9.5 | 1.5±0.5 | 11.6±4.4 | 1.9±0.6 | 3.6±2.4 | 60.2±9.0 |
| | LX | 8.4±2.7 | 3.0±1.3 | 17.1±15.4 | 24.3±9.1 | 2.2±2.6 | 8.9±3.9 | 1.4±0.4 | -0.1±2.1 | 62.4±10.3 |
| | MEK | 5.3±2.3 | 2.2±1.1 | 13.2±4.0 | 16.6±6.3 | 2.5±0.9 | 8.6±4.4 | 1.1±0.3 | -0.2±3.7 | 36.1±11.0 |
| | HY | 8.4±3.8 | 3.0±1.6 | 21.5±11.3 | 18.9±10.2 | 2.5±0.7 | 6.7±4.9 | 2.1±1.5 | -6.5±6.8 | 42.8±21.8 |