# Peer review of "Measurement report: Contrasting elevation-dependent changes in light absorption of black and brown carbon: lessons from *in-situ* measurements from highly polluted Sichuan Basin to pristine Tibetan Plateau"

_Atmospheric Chemistry and Physics, 2022_

## Referee Comment (RC1)

Review of "Measurement report: The first in-situ PM1 chemical measurements at the steep slope from highly polluted Sichuan Basin to pristine Tibetan Plateau: light absorption of carbonaceous aerosols, and source and origin impacts" by Zhao et al.

Recommendation: Minor Revisions

The manuscript presented the observational results of the first in-situ measurement of atmospheric aerosols, especially aerosol absorption properties, at six sites along eastern slope of the Tibetan Plateau. In general, the paper is well written and presented in a logical way. It is a timely and important piece of work, and of general interest for Tibetan Plateau and atmospheric aerosol related communities. I therefore recommend publication of this paper in Atmospheric Chemistry and Physics after minor revisions. My comments are listed as follows:

**Major Comments:**

Using Equation (5) to separate total aerosol absorption into EC and BrC absorption is an objective and effective method. But the shortcomings of the method should be kept in mind when analyzing and discussing the results. For example, the method does not consider the absorption of mineral dust (or fine soils), which accounts for very small percentage for most urban sites but might account for a large proportion of total aerosols for some other sites. Previous studies have revealed that the mineral dust is an important species of the atmospheric aerosols over the Tibetan Plateau (e.g., Zhang et al., 2021). Besides, assuming AAE of EC as 1 does not take into the aging of EC.

**Specific Comments:**

1. Page 5, Line 5: Is the meteorological data available for each site? Are the sampling sites near the meteorological observation sites?

2. Page 6, Line 21: "These are wavelength independent factors." Revise this sentence since it might be misleading.

3.  Page 6, Equation 5: Separating total aerosol absorption into EC and BrC absorption is applicable for urban sites with severe anthropogenic pollution and little mineral dust (fine soils).

4.  Page 6, Equation 5: Assuming AAE of EC as 1 excludes the influence of EC aging, which causes higher AAE than 1.

5.  Page 7, Line 6: The assumption of no vertical gradients within the PBL might overestimate the radiative forcing of aerosols.

6.  Page 7, Line 12: Why choose 405 nm as the lower limit of the integral?

7.  Section 2.5: Which version of the EPA PMF model was used in the study?

8.  Table 1: The abbreviations of the site names were not defined in the manuscript.

9.  Page 12 and Figure 7: The physical meaning of the parameter (radiative forcing of BrC relative to EC) is recommended to be further discussed. Were the nighttime samples used when calculating this parameter?

10. Figure 9: Black lines and circles are recommended. It is not necessary to use too many colors in this figure.

11. Page 16, Line 14: Delete "full".

---

## Author Response (AR1)

**Response to RC1**

The manuscript presented the observational results of the first in-situ measurement of atmospheric aerosols, especially aerosol absorption properties, at six sites along eastern slope of the Tibetan Plateau. In general, the paper is well written and presented in a logical way. It is a timely and important piece of work, and of general interest for Tibetan Plateau and atmospheric aerosol related communities. I therefore recommend publication of this paper in Atmospheric Chemistry and Physics after minor revisions.

Thank you for your positive comments on our manuscript (Title: Measurement report: The first in-situ PM1 chemical measurements at the steep slope from highly polluted Sichuan Basin to pristine Tibetan Plateau: light absorption of carbonaceous aerosols, and source and origin impacts, ID: acp-2022-241). Your constructive suggestions are very valuable for improving the quality of our manuscript. The corresponding revisions will be conducted in the new manuscript. The responses to the comments are given in detail as follows.

1) Major Comments: Using Equation (5) to separate total aerosol absorption into EC and BrC absorption is an objective and effective method. But the shortcomings of the method should be kept in mind when analyzing and discussing the results. For example, the method does not consider the absorption of mineral dust (or fine soils), which accounts for very small percentage for most urban sites but might account for a large proportion of total aerosols for some other sites. Previous studies have revealed that the mineral dust is an important species of the atmospheric aerosols over the Tibetan Plateau (e.g., Zhang et al., 2021). Besides, assuming AAE of EC as 1 does not take into the aging of EC.

**Response:** Thank you for your suggestions and providing the important recent study on mineral dust over the TP (Zhang et al., 2021). As you said, the main shortcoming of the separation of total aerosol absorption into EC and BrC (Eq. 5) is lack of considering the mineral dust impacts. According to the recent study of Zhang et al. (2021), mineral dust may be an important species of the atmospheric aerosols over the Tibetan Plateau. However, the study region is located at the eastern slope of TP during our campaign, which is more easily affected by anthropogenic sources from heavy polluted Sichuan Basin than natural sources such as mineral dust as compared to the north areas close to Taklimakan and Gobi Deserts. One main aim of this study is to reveal the gradient distributions of aerosol optical properties from the pollution Sichuan Basin to eastern TP, and thus the impact of the shortcoming may be less when studying the spatial heterogeneity of aerosol optical properties at relatively small spatial scale. In addition, AAE of EC is assumed as 1, and the aging of EC did not take when separating the total aerosol absorption into EC and BrC (Eq. 5) in our study. The above explanations will be added and the recent study will be cited in the revised manuscript.

2)  Page 5, Line 5: Is the meteorological data available for each site? Are the sampling sites near the meteorological observation sites?

**Response:** Thank you for your question. The meteorological data (temperature, RH, wind speed and direction) from China Meteorological Data Service Center (CMDSC) is available for each sampling site. PM1 samples were collected near the meteorological observation sites, and thus the meteorological variables can represent the situation at the study region. The above statements will be added to the revised manuscript.

3)  Page 6, Line 21: "These are wavelength independent factors." Revise this sentence since it might be misleading.

**Response:** Thank you for your reminder. "These are wavelength independent factors." will be changed to "They do not change as the wavelength." in Line 21 of Page 6 of the revised version of our manuscript.

4)  Page 6, Equation 5: Separating total aerosol absorption into EC and BrC absorption is applicable for urban sites with severe anthropogenic pollution and little mineral dust (fine soils).

**Response:** Thank you for your suggestion. The shortcoming of the separating method cannot be omitted when it is used at the locations with more mineral dust. However, as the response to your Comment 1), the study region is located at the eastern slope of TP during our campaign, which is more easily affected by anthropogenic sources from heavy polluted Sichuan Basin than natural sources such as mineral dust (Yin et al., 2020) as compared to the north areas close to Taklimakan and Gobi Deserts. The explanations will be included in the revised manuscript.

5)  Page 6, Equation 5: Assuming AAE of EC as 1 excludes the influence of EC aging, which causes higher AAE than 1.

**Response:** Thank you for your suggestion. AAE of EC is assumed as 1, and the aging of EC did not take when separating the total aerosol absorption into EC and BrC (Eq. 5) in our study. The above statements will be added to the revised manuscript.

6)  Page 7, Line 6: The assumption of no vertical gradients within the PBL might overestimate the radiative forcing of aerosols.

**Response:** Thank you for your suggestion. We assumed no vertical gradients of aerosols within the PBL. The assumption might overestimate the radiative forcing of aerosols, while it has less effect on the radiative forcing of BrC relative to EC (f).

The latter is more important for our study. The corresponding explanations will be given in the revised version of our manuscript.

7) Page 7, Line 12: Why choose 405 nm as the lower limit of the integral?

**Response:** Thank you for your question. The 405 nm is the lower limit of detection by the instrument of DRI-2015. Therefore, the radiative forcing of BrC relative to EC (f) is obtained by numerical integration of the above formula in the wavelength range of 405-980 nm and 405-445 nm for each sample. The explanation will be added to the revised manuscript.

8) Section 2.5: Which version of the EPA PMF model was used in the study?

**Response:** Thank you for catching that. EPA PMF 5.0 was used to apportion the sources in this study, which will be revised in the new manuscript.

9) Table 1: The abbreviations of the site names were not defined in the manuscript.

**Response:** Thank you for your reminder. In Table 1, Chengdu, Sanbacun, Wenchuan, Lixian, Maerkang and Hongyuan are abbreviated as CD, SBC, WC, LX, MEK and HY, respectively. The definition will be given in the table caption in the revised manuscript.

10) Page 12 and Figure 7: The physical meaning of the parameter (radiative forcing of BrC relative to EC) is recommended to be further discussed. Were the nighttime samples used when calculating this parameter?

**Response:** Thank you for your suggestions. The nighttime samples were excluded when calculating the radiative forcing of BrC relative to EC. The parameter (radiative forcing of BrC relative to EC) reflects light absorption strength of BrC at the shorter wavelengths as compared to that of EC aerosols at the whole wavelengths. The much higher f values indicated that radiative forcing of BrC aerosols is much stronger for the similar EC radiate forcing, and thus this parameter can be used to better understand the radiative forcing of secondary aerosols relative to primary aerosols at a specific location. The above discussion will be added to Page 12 of the revised manuscript.

11) Figure 9: Black lines and circles are recommended. It is not necessary to use too many colors in this figure.

**Response:** Thank you for your suggestions. Figure 9 and the similar figures in the supplemental materials will be revised according to your recommendation in the revised manuscript.

12) Page 16, Line 14: Delete "full".

**Response:** Thank you for catching that. The "full" in Line 14 of Page 16 will be deleted in the revised version of our manuscript.

**References**

Tian, P. F., Zhang, L., Ma, J., Tang, K., Xu, L., Wang, Y., Cao, X., Liang, J., Ji, Y., Jiang, J. H., Yung, Y. L., and Zhang, R. Radiative absorption enhancement of dust mixed with anthropogenic pollution over East Asia, Atmospheric Chemistry and Physics, 2018, 18, 7815-7825.

Zhang, L., Tang, C., Huang, J., Du, T., Guan, X., Tian, P. F., Shi, J., Cao, X., Huang, Z., Guo, Q., Zhang, H., Wang, M., Zeng, H., Wang, F., Dolkar, P. Unexpected high absorption of atmospheric aerosols over a western Tibetan Plateau site in summer. Journal of Geophysical Research: Atmospheres, 2021, 126, e2020JD033286.

**Response to RC2**

The manuscript gave measurement results of the $PM_1$ aerosol measurement at the steep slope from the Sichuan basin to Tibetan Plateau. In general, the measurement results are interesting and important to the related researchers. However, the paper was not well written and lack of some logic. Some major revisions were necessary.

Thank you for your positive and constructive comments on our manuscript (Title: Measurement report: The first in-situ $PM_1$ chemical measurements at the steep slope from highly polluted Sichuan Basin to pristine Tibetan Plateau: light absorption of carbonaceous aerosols, and source and origin impacts, ID: acp-2022-241). Your constructive suggestions are very valuable for improving the quality of our manuscript. The corresponding revisions will be conducted in the new manuscript. The responses to the comments are given in detail as follows.

Major Comments:

1 This manuscript requires corrections in punctuation, spelling, formatting, and references. Some examples are given in minor comments.
Response: Thank you for your suggestions. The punctuation, spelling, formatting, and references throughout the manuscript will be checked carefully and corrected in the revised manuscript.

2 The title should be simplified and I can't get the main point of the manuscript from the title.
Response: Thank you for your suggestions. The title will be simplified as "Measurement report: Contrasting elevation-dependent changes in light absorption of black and brown carbon: lessons from *in-situ* measurements from highly polluted Sichuan Basin to pristine Tibetan Plateau". The main point of the manuscript can be obtained from the revised title.

3 The abstract should be rewritten. The author should give the highlights and potential scientific meaning of this paper, and thus some brief introduction should be removed.
Response: Thank you for your suggestions. The brief introduction will be removed, and the highlights and potential scientific meaning of this paper will be gave in the rewritten abstract of the revised version of our manuscript.

4 As noted on page 4 line 40, their goals are to understand EC or BrC light absorption difference between the highly polluted basins and clean TP and reveal the corresponding mechanisms and provide a basic data set for optimization of regional climate modeling. They give the PMF results of the measurement. However, the PMF results were not discussed in their results. And the discussion of the PMF should relate to the light absorption properties of aerosols.

Response: Thank you for your constructive suggestions. The PMF results indicated that the winter secondary nitrate is a main source inside the SCB, which is closely related to more frequent secondary formation of nitrate in response to the intensive mixing between the motor vehicle emissions and other primary pollutants trapped inside the basin by strong capping inversion (Feng et al., 2020). Additionally, high humidity inside the SCB facilities the secondary nitrate formation, and the average nitrogen oxidation ratio in Sichuan (average RH = 80%) is 3.1 times of that in winter Beijing (average RH = 27%) (Wang et al., 2021). EC aerosols from the intensive human activities inside the SCB are easily aged by coating the secondary formed nitrate in winter, which further induces the enhancement of basin EC light absorption. The latest studies of Zhang et al. (2022) found that light absorption and radiative forcing of black carbon coated by inorganic salts are much stronger than that inside organic materials. In addition, the primary BrC from biomass burning and coal combustion for winter heating over the TP may partly contribute to the strong TP BrC light absorption. The above discussion on the relation of PMF results and the light absorption properties of aerosols will be included in the revised version of our manuscript.

5 The uncertainties of the method of deriving the BC, BrC, and MAC should be discussed.

Response: Thank you for your suggestions. EC and BrC were derived from light absorption coefficient ($b_{abs}$) depending on transmittance attenuation. For the seven-wavelength carbon analyzer, the filter transmittance ($FR_{\lambda}$, fraction of light transmitted through the filter) uncertainties range from 5% to 18%, with the best precision shown at 450 nm and 808 nm (Chen et al., 2015). The uncertainty is attributed to the quality of the laser and the sensitivity of the photodiode detector for different wavelengths.

The mass absorption efficiency (MAE) was obtained by the ratio of light absorption coefficients ($b_{abs,\lambda,EC}$ or $b_{abs,\lambda,BrC}$) to the corresponding EC or OC mass concentrations (Olson et al., 2015). The estimated $MAE_{BrC}$ was much lower than the true value by replacing BrC with OC due to BrC accounting for only a small fraction of OC. The main shortcoming of the separation of total aerosol absorption into EC and BrC (Eq. 5) is lack of considering the mineral dust impacts. According to the recent study of Zhang et al. (2021), mineral dust may be an important species of the atmospheric aerosols over the Tibetan Plateau. However, the study region is located at the eastern slope of TP during our campaign, which is more easily affected by anthropogenic sources from heavy polluted Sichuan Basin than natural sources such as mineral dust (Yin et al., 2020) as compared to the north areas close to Taklimakan and Gobi Deserts. One main aim of this study is to reveal the gradient distributions of aerosol optical properties from the pollution Sichuan Basin to eastern TP, and thus the impact of the shortcoming may be less when studying the spatial heterogeneity of aerosol optical properties at relatively small spatial scale. In addition, AAE of EC is assumed as 1, and the aging of EC did not take

when separating the total aerosol absorption into EC and BrC (Eq. 5) in our study. The above discussion will be included in the revised manuscript.

6 When considering the primary definition of $MAE_{EC}$, it is determined by the core size distribution (emission sources), refractive index (coating chemical components), coating thickness (aging scale of BC), and some other factors. The author gives the relationship of $MAE_{EC}$ with many inorganic components in figure 6, please give the reason why they also relate the MAE with these factors.

Response: Thank you for your suggestions. As you said, $MAE_{EC}$ is determined by the core size distribution, refractive index, coating thickness and some other factors, which can represent emission sources, coating chemical components and aging scale of EC. A specific inorganic component can be considered as the indicator of the specific emission sources. $K^+$ and $Cl^-$ ions are usually used for characterizing biomass burning (BB) and coal combustion (CC), respectively (Tao et al., 2016). $NO_3^-$ and $SO_4^{2-}$ can reflect motor vehicle and industry source impacts, respectively. Therefore, Figure 6 gives the relationship of $MAE_{EC}$ with the concentrations of $K^+$, $Cl^-$, $NO_3^-$ and $SO_4^{2-}$ ions to better find key sources impacting $MAE_{EC}$. The explanations will be added to the revised version of our manuscript.

7 In section 3.3, the author gives Regional and long-range transport impacts discussion. How were their results related to the previous sections?

Response: Thank you for your constructive suggestions. As you might think, the discussion on regional and long-range transport impacts should be related to the light absorption of aerosols. The fresh aerosol particles are gradually aged during the long-range transport, and then enhances the light absorption and radiative forcing of aerosols. The coefficients of variation (CV) between the basin and plateau sites for the mean mass concentrations of water-soluble ions and carbonaceous species (Figs 9 and S10-S12) indicated that the air pollutants weakly interact between western SCB and eastern TP. Therefore, the light absorption of carbonaceous aerosols over the TP is less influenced by pollutants from the SCB. The weak correlations for $Na^+$, $Mg^{2+}$, and $Ca^{2+}$ values in spring and winter between basin and plateau sites may suggest local and regional dust plume impacts. Therefore, the lack of considering mineral dust impacts in separation of BrC light absorption from total aerosol absorption (Eq. 5) might cause some errors. The errors should be much smaller as compared to north or northeast TP close to Taklimakan and Gobi Deserts.

The back trajectories suggested that the biomass burning emissions originated from South Asia can be transported to the eastern TP. The BrC aerosols from the intensive biomass burning in South Asia are gradually aged by internal or external mixing with the other anthropogenic emissions during the long-range transport. The light absorption of the aged BrC aerosols over the TP is enhanced by coating the inorganic components (Zhang et al., 2022), which may partly contribute much stronger BrC light absorption at the plateau sites than the basin sites. Unlike the eastern TP, the carbonaceous aerosols in western SCB are

regionally transported from the central and eastern SCB, which can be seen from pollution rose and back trajectories. The aerosols accumulate and stagnate at the front areas of the mountains due to terrain block, and thus the light absorption of EC aerosols emitted from motor vehicles is enhanced by the intensive mixing among the air pollutants.

Minor Comments:

1 Page 1: line 19: the "third pole", line 21 less: fewer, line26 on the east side of
Response: Thank you for your careful checking. The spelling will be revised according to your suggestions in the revised manuscript.

2 Page 8, line 25: this line should be in the introduction part.
Response: Thank you for your catching that. The sentence in Line 25 of Page 8 will be moved to the introduction part of the revised manuscript.

3 A table that summarizes the measurement sites (name, location, altitude) should be included in the manuscript.
Response: Thank you for your suggestions. A table that summarizes the measurement sites (name, location, altitude) will be included in the revised manuscript.

Table 1 Summary of the measurement sites (name, location and altitude).

| Name | Latitude (degree) | Longitude (degree) | Altitude (km) |
|---|---|---|---|
| Chengdu | 30.67 | 104.06 | 0.50 |
| Sanbacun | 30.99 | 103.66 | 0.65 |
| Wenchuan | 31.46 | 103.61 | 1.33 |
| Lixian | 31.42 | 103.16 | 1.89 |
| Maerkang | 31.92 | 102.22 | 2.62 |
| Hongyuan | 32.79 | 102.55 | 3.50 |

4 Page 9, line 4: the explanations of the higher OC/EC ratios with the altitude were not convincing. It can also result from stronger EC emissions at lower altitudes.
Response: Thank you for your suggestions. The much higher OC/EC ratios at the plateau sites than that at the basin sites may suggests that more secondary OC is formed by chemical reactions over Tibetan Plateau, which can be supported by the works of Wu et al. (2018). However, as you said, the higher OC/EC ratios with the altitude can also result from stronger EC emissions at lower altitudes. The above statements will be added to the revised manuscript.

5 Figure 1: the text and location are not clear, it should be reorganized.

Response: Thank you for catching that. The text and location of the measurement sites will be reorganized in the revised Figure 1 in the revised manuscript.

[Figure]

**Figure 1:** Geographic location of the six *in-situ* observation sites (Chengdu, Sanbacun, Wenchuan, Lixian, Maerkang, and Hongyuan) along the eastern slope of Tibetan Plateau. The map is a pure reproduction of Google Maps with added a marks for our study locations. Copyright © Google Maps.

6 Figure 2: EC fraction should be EC absorption. BC fraction should be BrC absorption.

Response: Thank you for your reminder. EC fraction and BrC fraction in Figure 2 will be changed to EC absorption and BrC absorption in the new Figure 2 in the revised manuscript.

[Figure]

**Figure 2:** Spectral light absorption coefficients ($b_{abs}$) by EC and BrC in spring and winter at the six sites along the eastern slope of Tibetan Plateau. The subplots depict the decomposition of total light absorption by EC and BrC with the model given in Eq. 4. Error bars represent uncertainties derived from replicate analyses and lower quantifiable limits.

7 Figure 4: The $R^2$ is not described here. Why not give the $R^2$ of spring in figure 4(b)?

Response: Thank you for your suggestion. Figure 4 shows $MAE_{BrC}$ and $MAE_{EC}$ variations as altitude in spring and winter during the campaign. The relationships between averaged MAE and altitude of the measurement sites were fitted by exponential function, and coefficients of determination ($R^2$) were gave in the figure. $R^2$ can reflect the strength of the relationships between two parameters. The contrasting MAE variations as altitude between BrC and EC in winter ($R^2$ is 0.89

for $MAE_{BrC}$ and 0.86 for $MAE_{EC}$) are more significant than those in spring ($R^2$ is 0.45 for $MAE_{BrC}$ and 0.06 for $MAE_{EC}$). The better relationships in winter may be because more urban and aged sources are trapped inside the deep basin in response to strong temperature inversion in winter (Feng et al., 2020).

[Figure]

**Figure 4:** Variations of (a) $MAE_{BrC}$ and (b) $MAE_{EC}$ at 405 nm as altitude in spring and winter during the campaign. The solid dots denote the median values; the two whiskers of the dots denote the 25th and 75th percentiles. The relationships between averaged MAE and altitude of the measurement sites were fitted by exponential function, and the coefficients of determination ($R^2$) also were given in each subplot. The relationships ($R^2$ with the superscript of an asterisk) passed the significance level of 0.01.

8 Figure 7: why did the author give the $R^2$ here?
Response: Thank you for your reminder. $R^2$ in Figure 7a was deleted (see the below figure).

[Figure]

**Figure 7:** Variation of radiative forcing of BrC relative to EC (f, see Eq. 8) as (a) altitude and (b) OC/EC ratio for each site. The solid dots denote the median values; the two whiskers of the dots denote the 25th and 75th percentiles of the variables.

9 Figure 10 should be placed in the supplement.

Response: Thank you for catching that. Figure 10 will be moved to the supplement.

**References**

Chen, L. W. A., Chow, J. C., Wang, X. L., Robles, J. A., Sumlin, B. J., Lowenthal, D. H., Zimmermann, R., and Watson, J. G.: Multi-wavelength optical measurement to enhance thermal/optical analysis for carbonaceous aerosol, Atmospheric Measurement Techniques, 8(1), 451–461, 2015.

Feng, X. Y, Wei, S. M, and Wang, S. G.: Temperature inversions in the atmospheric boundary layer and lower troposphere over the Sichuan Basin, China: Climatology and impacts on air pollution, Science of the Total Environment, 726, 138579, 2020.

Tao, J., Zhang, L. M., Zhang, R. J., Wu, Y. F., Zhang, Z. S., Zhang, X. L., Tang, Y. X., Cao, J. J., and Zhang, Y. H.: Uncertainty assessment of source attribution of PM2.5 and its water-soluble organic carbon content using different biomass burning tracers in positive matrix factorization analysis-a case study in Beijing, China, Science of the Total Environment, 543, 326–335, 2016.

Wang, Y. J., Hu, M., Hu, W., Zheng, J., Niu, H. Y., Fang, X., Xu, N., Wu, Z. J., Guo, S., Wu, Y. S., Chen, W. T., Lu, S. H., Shao, M., Xie, S. D., Luo, B., and Zhang, Y. H.: Secondary formation of aerosols under typical high-humidity conditions in wintertime Sichuan Basin, China: A contrast to the North China Plain, Journal of Geophysical Research: Atmospheres, 126(10), e2021JD034560. https://doi.org/10.1029/2021JD03456, 2021.

Wu, G. M., Wan, X., Gao, S. P., Fu, P. Q., Yin, Y. G., Li, G., Zhang, G. S., Kang, S. C., Ram, K., and Cong, Z. Y.: Humic-like substances (HULIS) in aerosols of central Tibetan Plateau (Nam Co, 4730 m asl): abundance, light absorption properties, and sources, Environmental Science and Technology, 52(13), 7203−7211, 2018.

Yin, D. Y., Zhao, S. P., Qu, J. J., Yu, Y., Kang, S. C., Ren, X. L., Zhang, J., Zou, Y., Dong, L. X., Li, J. L., He, J. J., Li, P., and Qin, D. H.: The vertical profiles of carbonaceous aerosols and key influencing factors during wintertime over western Sichuan Basin, China, Atmospheric Environment, 223, 117269, 2020.

Zhang, J., Wang, Y. Y., Teng, X. M., Liu, L., Xu, Y. S., Ren, L. H., Shi, Z. B., Zhang, Y., Jiang, J. K., Liu, D. T., Hu, M., Shao, L. Y., Chen, J. M., Martin, S. T., Zhang, X. Y., and Li, W. J.: Liquid-liquid phase separation reduces radiative absorption by aged black carbon aerosols, Communications Earth & Environment, 3, 128, https://doi.org/10.1038/s43247-022-00462-1, 2022.

Zhang, L., Tang, C. G., Huang, J. P., Du, T., Guan, X., Tian, P. F., Shi, J. S., Cao, X. J., Huang, Z. W., Guo, Q., Zhang, H. T., Wang, M., Zeng, H. Y., Wang, F. Y., and Dolkar, P.: Unexpected high absorption of atmospheric aerosols over a western Tibetan Plateau site in summer, Journal of Geophysical Research: Atmospheres, 126(7), e2020JD033286. https://doi.org/10.1029/2020JD033286, 2021.

---

## Editor Decision (ED1)

Editor's review for acp-2022-214

Examples of phrases with unclear text and missing articles/prepositions:

Abstract

Line 18: Replace "of aerosols" by "by aerosols"

Line 20: Replace "extending elevation" by "ranging in elevation

Lines 21-22: The phrase "The light absorption of brown carbon (BrC) accounting for that of total carbon increases from 20%..." is unclear in English.  I suggest your replace it by: " The fraction of light absorption by brown carbon (BrC) to total carbon increases from 20%..."

Lines 23-24: "Contrary to BrC aerosols, winter EC (elemental carbon) mass absorption efficiency declines with altitude"  is an unclear sentence. I suggest the following text " In contrast, the mass absorption efficiency elemental carbon (EC) in winter decreases with altitude"

Line 25: Insert: "Emissions from the more..." at the beginning of the sentence,  since the sources are not transported, it is the emissions.

Line 26: Replace: "to winter stable air inside.." by "to the stable air in winter inside.."

Line 26:  Replace "which also is" by "which is also"

Lines 29-31: The phrase "and thus the enhanced radiative forcing of BrC relative to EC from polluted SCB to pristine TP is because the concentration of

OC decreases more slowly with altitude than does EC." is unclear. I recommend that you re-write is as a separate sentence as "Thus, the reason of the enhanced relative BrC to EC radiative forcing from polluted SCB to pristine TP is that the BrC concentration decreases more slowly than the EC concentration with altitude"

Lines 31-33: Rewrite this last sentence for clarity, suggestion:  "This study contributes to the understanding of the difference in light absorption by EC and BrC with altitude, from polluted lower-altitude basins to the pristine TP, and provides a data set for regional climate model validation."

---

## Author Response (AR2)

Dear Prof. Raga,

Thank you for your good suggestions. The manuscript (ID: ACP-2022-241) was copyedited by a professional copyediting service according to the suggestions. Certificate is given as follows.

[Figure]

**Certificate of Elsevier Language Editing Services**

The following article was edited by Elsevier Language Editing Services:

"Measurement report: Contrasting elevation-dependent changes in light absorption of black and brown carbon: lessons from in-situ measurements from highly polluted Sichuan Basin to pristine Tibetan Plateau"

Authored by:

**Suping Zhao**

Date: 15-Oct-2022
Serial number: LE-250964-533BCDC8B9B9